# Efficacy of hyperthermic intraperitoneal chemotherapy in colorectal cancer: A phase I and III open label randomized controlled registry-based clinical trial protocol

**Lana Ghanipour**[1,2]*, **Gabriella Jansson Palmer**[3], **Per J. Nilsson**[3], **Caroline Nordenvall**[3], **Jan-Erik Frödin**[4], **Elinor Bexe Lindskog**[5], **Dan Asplund**[5], **Torbjörn Swartling**[5], **Wilhelm Graf**[1,2], **Helgi Birgisson**[1,2], **Ingvar Syk**[6], **Victor Verwaal**[6], **Jenny Brändstedt**[6], **Peter H. Cashin**[1,2]

1 Uppsala University, Uppsala, Sweden, 2 Department of Surgical Sciences, Gastro Intestinal Surgery, Uppsala University Hospital, Uppsala University, Uppsala, Sweden, 3 Department of Pelvic Cancer, GI Oncology and Colorectal Surgery Unit, Karolinska University Hospital, Stockholm, Sweden, 4 Department of Oncology-Pathology, Karolinska University Hospital, Stockholm, Sweden, 5 Department of Surgery, Institute of Clinical Sciences, Sahlgrenska University Hospital, Gothenburg, Sweden, 6 Department of Clinical Sciences, Lund University, Skåne University Hospital, Malmö, Sweden

* lana.ghanipour@surgsci.uu.se

**Data Availability Statement:** No datasets were generated or analyzed during the current study. All

## Abstract

Standard treatment for patient with peritoneal metastases from colorectal cancer is cytoreductive surgery (CRS) and hyperthermic intraperitoneal chemotherapy (HIPEC). In recent years, the efficacy of oxaliplatin-based HIPEC has been challenged. An intensified HIPEC (oxaliplatin+irinotecan) in combination with early postoperative intraperitoneal chemotherapy (EPIC) has shown increased recurrence-free survival in retrospective studies. The aim of this trial is to develop a new HIPEC/EPIC regimen and evaluate its effect on morbidity, oncological outcome, and quality-of-life (QoL). This study is designed as a combined phase I/III multicenter randomized trial (RCT) of patients with peritoneal metastases from colorectal cancer eligible for CRS-HIPEC. An initial phase I dose escalation study, designed as a 3 +3 stepwise escalation, will determine the maximum tolerable dose of 5-Fluorouracil (5-FU) as 1-day EPIC, enrolling a total of 15–30 patients in 5 dose levels. In the phase III efficacy study, patients are randomly assigned intraoperatively to either the standard treatment with oxaliplatin HIPEC (control arm) or oxaliplatin/irinotecan-HIPEC in combination with single dose of 1-day 5-FU EPIC (experimental arm). 5-FU is administered intraoperatively after CRS-HIPEC and closure of the abdomen. The primary endpoint is 12-month recurrence-free survival. Secondary endpoints include 5-year overall survival, 5-year recurrence-free survival (registry based), postoperative complications, and QoL up to 3 years after study treatment. This phase I/III trial aims to identify a more effective treatment of colorectal peritoneal metastases by combination of HIPEC and EPIC.

relevant data from this study will be made available upon study completion.

**Funding:** Complete funding has been achieved through the Swedish Research Council. The funders had no role in study design, data collection and analysis, decision to publish, or preparation of the manuscript. None of the authors have received any salary from the funder.

**Competing interests:** No authors have competing interests.

# Introduction

Colorectal cancer with peritoneal metastases (CRCPM) is now routinely treated with cytoreductive surgery (CRS) and hyperthermic intraperitoneal chemotherapy (HIPEC). The PRODIGE 7 trial, has however cast doubt on the efficacy of the 30-minute single-drug oxaliplatin HIPEC for CRCPM [1,2]. Due to several methodological limitations in the PRODIGE 7 study, the role and effect of HIPEC raises many questions and incentivizes the need for further research. However, there are randomized trials with proof of concept from both gastric cancer and ovarian cancer showing that HIPEC provides a survival benefit for patients treated with CRS [3,4]. Thus, it seems clinically relevant and meaningful to continue the search for a HIPEC regimen that is effective in colorectal cancer. In two recent studies, a disease-free survival benefit in favor of oxaliplatin/irinotecan HIPEC vs single drug oxaliplatin HIPEC was demonstrated [5,6]. Moreover, several studies have investigated intensified HIPEC treatment using two drugs and also by adding early postoperative intraperitoneal chemotherapy (EPIC) [7–14]. Several trials have failed to prove any benefit of oxaliplatin based HIPEC, leading to several reviews questioning the place for oxaliplatin in HIPEC [15,16]. It can be argued that a 30-minute single-drug oxaliplatin HIPEC might be insufficient in preventing peritoneal relapse or increasing overall survival. Thus, the purpose of this clinical phase I and III trial is to develop and evaluate a new intensified HIPEC regimen based on a combination of HIPEC and EPIC.

# Materials and methods

## Basic design and setting

This clinical trial is divided into two parts—phase I and III. The phase I trial will determine the maximum tolerated dose of a single 5-Fluorouracil dose administered as early postoperative intraperitoneal chemotherapy (EPIC) following an intensified HIPEC, by the combination of oxaliplatin and irinotecan. It will be designed as a 3 + 3 dose titration study, with dose escalation in five steps, up to 850 mg/m$^2$.

The phase III part will randomize patients to either the current Swedish standard treatment of single-drug oxaliplatin HIPEC (460 mg/m$^2$) *vs.* intensified oxaliplatin (360 mg/m$^2$) and irinotecan (360 mg/m$^2$) HIPEC combined with 1 day of 5-fluorouracil (5-FU) EPIC. The dose of 5-FU will be established following the results of the phase- I trial. The intraabdominal treatment is preceded by intravenous bolus injection of 5-FU (400 mg/m$^2$) and calcium folinate (60 mg/m$^2$) both in the standard and the experimental treatment arm. The trial will be conducted in all four HIPEC centers performing HIPEC treatment in Sweden. International centers are still pending. The primary endpoint of the phase III trial is the 12-month recurrence-free survival.

## Sample size and inclusion and exclusion criteria

The sample size of the phase I study will be 15–30 patients depending on how many of the five dose levels require 6 patients. Regarding the phase III study, the power calculation was based on an alpha of 5% and beta of 80% and an expected 12-month recurrence free survival benefit from 50% till 66%, giving 147 patients needed in each arm. With 10% loss to follow up, 163 patients in each arm will be needed. Hence, in total, 326 patients are to be included which will define the sample size. An interim analysis is planned in the phase III part after 140 patients (70 in each arm) have been included.

SUBJECT INCLUSION CRITERIA

1. Provision of written informed consent prior to any study specific procedures.

2. ECOG Performance Status Score 0,1 or 2 alternatively Karnofsky 60–100.

3. Adequate kidney, liver, bone marrow function according to laboratory tests. The laboratory test needs to be normal according to reference values or within +/- 20% of the highest respective lowest value.

4. For females of childbearing potential, a negative pregnancy test must be documented.

5. ≥ 18 years old and <75 years old.

6. Colorectal cancer with peritoneal metastases with or without liver metastases. The liver metastases must to have been removed prior to CRS-HIPEC or synchronously with the CRS-HIPEC. Only patients requiring non-complex liver resections of not more than 3 metastases are eligible.

7. All patients deemed eligible for CRS and HIPEC according to clinical routine.

SUBJECT EXCLUSION CRITERIA

1. Previous severe toxicity/allergic reactions to systemic chemotherapy agents oxaliplatin or irinotecan or 5-FU.

2. Unable to tolerate intensified HIPEC treatment due to comorbidity.

3. Metastasis other than peritoneum or liver.

4. Need for complex liver-perenchymal sparing surgery or hemihepatectomy procedures.

5. Previous CRS or HIPEC.

6. Pregnant or lactating (nursing) women.

7. Active infections requiring antibiotics.

8. Active liver disease with positive serology for active hepatitis B, C, or known HIV.

9. Concurrent administration of any cancer therapy other than planned study treatment within 4 weeks prior to and up to 4 weeks after study treatment.

10. Incomplete cytoreduction defined as completeness of cytoreduction score 1–3.

11. Histopathology of other origin than colorectal cancer.

## Design details phase I

In the dose escalation phase I trial (Table 1), the titration groups (consisting of 3 or 6 patients) are followed for 30 days postoperatively after which the Data Monitoring Committee (DMC) will determine whether to increase the 5-FU dose for the following group of patients or not.

A maximum of 33% Clavien Dindo grade 3b-5 complications defines the limit of tolerance. All patients will be reviewed at each of the dose levels by the DMC prior to allowing the next group of patients to advance to the following level. The toxicity follow-up is 30 days if no incidents occur. If the postoperative rehabilitation is slower or incidents occur, the follow-up period may be extended by the site Principal Investigator or Co-investigator.

If none or one of the first three patients at a dose level has a grade 3b-4b complication the DMC will review the data and permit the start of the next level. If more than one patient experiences a grade 3b-4b complication another three patients will be included at the same dose level before the DMC will review the data. A frequency of grade 3b-4b morbidity in ≤1 of 3 or

**Table 1. Dose escalation of 5-FU 24-hour EPIC for each titration level and concomitant treatment where irinotecan is also an investigational drug (the other drugs are non-investigational and given to both groups in the randomized part of the trial).**

| Level | Non-investigational Oxaliplatin | Investigational Irinotecan | Investigational 5-FU 24-hour EPIC | Non-investigational 5-FU bolus | Non-investigational Calcium folinate |
|---|---|---|---|---|---|
| -1 | 360 mg/m$^2$ | **360 mg/m$^2$** | **250 mg/m$^2$** | 400 mg/m$^2$ | 60mg/m$^2$ |
| 0 | 360 mg/m$^2$ | **360 mg/m$^2$** | **400 mg/m$^2$** | 400 mg/m$^2$ | 60mg/m$^2$ |
| 1 | 360 mg/m$^2$ | **360 mg/m$^2$** | **550 mg/m$^2$** | 400 mg/m$^2$ | 60mg/m$^2$ |
| 2 | 360 mg/m$^2$ | **360 mg/m$^2$** | **700 mg/m$^2$** | 400 mg/m$^2$ | 60mg/m$^2$ |
| 3 | 360 mg/m$^2$ | **360 mg/m$^2$** | **850 mg/m$^2$** | 400 mg/m$^2$ | 60mg/m$^2$ |

$\leq$2 of 6 patients may result in a recommendation by the DMC to proceed to the next dose level and this defines the dose-limiting toxicity. A higher frequency of grade 3b-4b complications means the maximum tolerated dose (MTD) has been passed, and the dosage used in the efficacy study will be the nearest underlying dose level. Any grade 5 event pauses all inclusion for review. Depending on cause, the trial may continue after DMC review.

## Design phase III

To study efficacy in the phase III part, an open label randomized clinical trial will commence after the DMC has determined the maximum tolerated dose of 5-FU EPIC. Treatments used in the trial are outlined below. The randomization will be intraoperatively at a 1:1 ratio. Cytoreductive procedures will be the same in both arms. The trial allows for dose reduction at the surgeon's or the oncologist discretion; either full dose is given, or a 25% dose reduction as defined below.

<u>Arm A (HIPEC) Standard treatment (normal dose- (25% dose level reduction))</u>

- Oxaliplatin– 460 mg/m$^2$ (HIPEC); (350 mg/m$^2$).

- 5-FU– 400 mg/m$^2$ IV bolus (Intraop)–(300 mg/m$^2$).

- Calcium folinate 60 mg/m$^2$ IV (Intraop).

- Granulocyte colony stimulating factor given postoperative only when indicated by neutrophil count drop below reference value.

<u>Arm B (HIPEC + EPIC) Experimental treatment (25% dose reduction)</u>

- Oxaliplatin– 360 mg/m$^2$ (HIPEC)–(270 mg/m$^2$).

- Irinotecan– 360 mg/m$^2$ (HIPEC)–(270 mg/m$^2$).

- 5-FU– 400 mg/m$^2$ IV bolus (Intraop)– (300 mg/m$^2$).

- Calcium folinate– 60 mg/m$^2$ IV (Intraop).

- 5-FU– 250–850 mg/m$^2$ (190–640 mg/m$^2$) IP, diluted in 400ml of 9% saline for a day (End of procedure).

  ○ The EPIC treatment is administered in the operating theatre after the abdomen is completely closed. The dose will be divided equally into 2 injections of 200ml and given through two separate abdominal drains. The drains may be flushed with 20ml saline solution to prevent retention of 5-FU in the drain catheter. Afterwards, all drains are clamped and kept so for a day to evaluate remaining fluid from the abdominal cavity, drains are opened the day after end of the procedure.

- Granulocyte colony stimulating factor (G-CSF–e.g. filgrastim 5μg/kg/day) is administered prophylactically on postoperative day 4 to 8 (5 days). If the neutrophil count is below reference value after 5 days of prophylactic treatment, the treatment may continue until the count is normalized, as clinically indicated. If leucocytosis occurs, the prophylactic treatment can be terminated earlier than after 5 days.

G-CSF may be administered as needed before or after the prophylactic period if an early or late neutropenia is detected.

## Outcomes

The primary outcome of the phase I part is complications according to Clavien-Dindo grading, whereas the primary outome of the phase III part is 12-month recurrence-free survival. Type of recurrences (peritoneum, liver, lung or lymph-nodes) will be recorded. Normal work-up and baseline variables will be collected as well as relevant surgical variables. Besides morbidity and recurrence-free survival, overall survival and quality of life will be followed up according to national guidelines.

## Follow up

Standard follow up after CRS-HIPEC will be performed with contrast-enhanced thoraco-abdominal CT scans every 6 months for 2 years and then annually until 5 years. Regular laboratory blood test with tumour markers (CEA, CA19-9 and CA 125) will be assessed according to standard clinical routine, as above schedule.

European organization for research and treatment of cancer (EORTC) QLQ-30, QLQ-CR29 and STO22 will be used to evaluate quality of life. Questionnaires are sent out at baseline and postoperatively at 3, 12, and 36 months. All patients are included in the quality-of-life registry as part of the clinical quality assurance follow up, unless they specifically have requested not to be included.

## Data management

The Swedish National HIPEC registry, established 2012, is a quality registry that includes data of preoperative work-up, surgical and oncological outcome in patients with peritoneal metastases.

All patients are registered preoperatively in the Swedish National HIPEC registry, at the preoperative outpatient clinic visit, when informed consent will be retrieved.

Data input will be reviewed by the monitors. Incomplete data, will be listed in a data clarification form to be sent to respective site to resolve inconsistencies and missing information. A copy will be returned to the sponsor. A complementary eCRF part will be added to the HIPEC registry to collect the remaining necessary information for the trial.

## Statistical analyses

For the phase I study part, no advanced statistical analyses are necessary. For the phase III part of the study, the interim safety analysis after 140 included patients will evaluate morbidity according to Clavien-Dindo (CD) grading. The study will be terminated if occurrence of morbidity exceeding 3 in the CD scale, in 40% or more of included patients. Furthermore, an overall survival analysis (minimum of 12 months follow-up) will be performed in the interim analysis in order to assess whether overall survival may be used as primary endpoint or not. There will be a minimum of 12 months follow-up.

The primary analysis for the phase III part will be a multivariable Cox regression analysis with 12-months recurrence free survival as endpoint including the following parameters: treatment arm, age, PCI-score (continuous variable), use of systemic perioperative chemotherapy, liver metastases resection, colon or rectal primary, and lymph-node metastasized primary tumor. The definition of recurrence free survival includes time from surgery to the date of any causes of death, intra- or extra-abdominal relapses, except for any second primary CRCs nor primary non-CRC. The primary analysis can commence once all patients have been observed for a minimum of 12 months. Further secondary analysis will include a multivariable logistical regression analysis with 12-month recurrence free survival as endpoint. A Fisher's exact test for 12-months recurrence free survival between the arms, a Kaplan-Meier curve with two-tailed log rank test between the arms with recurrence free survival up to 5 years as endpoint (i.e. time to recurrence) will be performed. The same analyses above may be run against peritoneal recurrence-free survival as secondary analyses as well.

If the interim analysis results demonstrate that overall survival is attainable as primary endpoint, a new statistical analysis will be set up after a renewed sample size calculation. Further analyses of secondary endpoints have been defined in the protocol.

Block randomization was chosen as the method of randomization. Patients with no liver metastases were randomized with center stratification and PCI stratification (1–10 vs 11+) with a block size of 6. Patients with liver metastases were randomized nationally with stratification of only PCI (1–10 vs 11+) also with a block size of 6.

## Ethical considerations

The study will be conducted in accordance with the protocol, applicable regulatory requirements, Good Clinical Practice (GCP) and the ethical principles of the Declaration of Helsinki as adopted by the 18th World Medical Assembly in Helsinki, Finland, in 1964 and subsequent versions.

The main ethical considerations in this research project are related to the balance of procedural morbidity/mortality and efficacy/benefit. CRS and HIPEC treatment is associated to a high risk for complications, thus it is important to monitor that the morbidity does not lead to increased severe postoperative morbidity or mortality. Furthermore, it is important that morbidity does not lead to increased long-term functional impairment. Out of 9 previous studies on EPIC, 3 studies indicated that there may be increased morbidity (but none for mortality) [7]. The other 6 studies showed no difference. It is not anticipated that this study will lead to a general increase in morbidity or mortality. However, this intensified HIPEC arm has not been followed up in a prospective systematic way. Neither has the maximum tolerated dose for a day 5-FU been determined. Therefore, this trial will begin as a phase I study that moves into a randomized open-label phase III study. In the interim analysis after half the patients in phase III part have been included, the morbidity results will be reviewed by the DMC before finishing the phase III step.

One specific morbidity that has been raised concerning intensified HIPEC treatment is neutropenia [11]. This is a specific complication that most likely will be increased with the intensified treatment. However, a recent study [8] investigating the specific effects of neutropenia showed that other serious side-effects were not increased nor mortality. Unexpectedly, neutropenia was a positive prognostic factor. Moreover, the effects of neutropenia are temporary and mitigated relatively easily with granulocyte-colony stimulating factor. Lastly, to monitor the patient's perspective on the side-effect/benefit balance, a quality-of-life part of the trial will be included to follow-up health-related quality-of-life. As seen in the safety review from an earlier study [7], the morbidity of adding EPIC treatment to HIPEC is uncertain. This will

alleviate potential increased risks with adding EPIC treatment to HIPEC. Considering the significant risk of relapse for this disease, the potential benefits outweigh the risks.

This trial is approved by the Swedish Ethical Review Authority, Dnr:2022-04332-02.

## Timeline

This phase I and III trial started with its' first inclusion in May of 2021 and is planned to be completed by 2025-12-31.

## Discussion

The EFFIPEC trial program aims at increasing the survival in patients with colorectal peritoneal metastases, by a new intensified HIPEC and EPIC regimen. In view of results reported from gastric and ovarian cancer, it stands to reason that there is hope for an effective HIPEC treatment for colorectal peritoneal metastases. The HIPEC treatment to be tested is designed to address two previous problems. First, the issue of chemotherapy resistance which may cause a reduced efficacy particularly in patients previously treated with oxaliplatin- based chemotherapy [17]. However, as seen in a recent publication, combining oxaliplatin and irinotecan might decrease risk of receiving a HIPEC treatment to which the patient's tumor is resistant [18]. Secondly, the problem of short exposure to chemotherapy has been rectified with the addition of 1-day EPIC. This EPIC dose will be dose titrated to maximize the impact of the longer exposure time in the abdomen.

There is a need for a new global standard for HIPEC treatment of colorectal peritoneal metastases. The results of the PRODIGE 7, COLOPEC, and Prophylochip trials have left us in a conundrum [1,15,16]. What is the actual current standard treatment? Some may say that CRS only is the standard, whereas more commonly, many centers have switched to mitomycin C despite lacking evidence [6].

There is one recently published trial demonstrating a locoregional effect from mitomycin C, the Spanish RCT HIPECT4, which aims to determine the effectiveness and safety of adjuvant HIPEC treatment with Mitomycin C in locally advanced primary colon cancer (cT4) without presence of peritoneal disease. The results of the trail, presented as abstract, have shown a significantly improved 3- years locoregional control rate of 97% in the HIPEC group vs 87% in the control group. However, no difference in DFS and OS were observed between the groups [19].

This trial has some limitations. The peritoneal surface oncology national guidelines do not recommend neoadjuvant chemotherapy primarily (other than in downstaging attempts), thus most patients will probably get upfront CRS with HIPEC. This may affect the generalizability of the results since many centers worldwide administer neoadjuvant chemotherapy as part of the standard treatment. However, as recently reported, it appears that neoadjuvant treatment does not have any clear benefit in this setting [20]. Furthermore, all four HIPEC centers in Sweden are joining this trial endeavor. As there is a difference in surgical volume, there may be a question of surgical experience between the HIPEC centers. However, yearly reports from the HIPEC registry and elaborated guidelines have consistently shown similar outcomes between the centers. Every other week the centers have a joint multidisciplinary conference to discuss the eligibility of certain cases on a national level. This has helped homogenize the selection process over time.

In conclusion, this is an important trial to complete as it may provide much needed answers as to the role of HIPEC in combination with CRS. We aim to establish a new global standardized HIPEC/EPIC regimen for colorectal cancer with peritoneal metastases.

## Supporting information

**S1 File.**
(DOCX)

**S2 File.**
(DOCX)

**S3 File.**
(DOCX)

## Author Contributions

**Conceptualization:** Lana Ghanipour, Gabriella Jansson Palmer, Per J. Nilsson, Caroline Nordenvall, Jan-Erik Frödin, Elinor Bexe Lindskog, Dan Asplund, Torbjörn Swartling, Helgi Birgisson, Ingvar Syk, Victor Verwaal, Jenny Brändstedt, Peter H. Cashin.

**Funding acquisition:** Peter H. Cashin.

**Methodology:** Lana Ghanipour, Gabriella Jansson Palmer, Per J. Nilsson, Caroline Nordenvall, Jan-Erik Frödin, Elinor Bexe Lindskog, Dan Asplund, Torbjörn Swartling, Wilhelm Graf, Helgi Birgisson, Ingvar Syk, Victor Verwaal, Jenny Brändstedt, Peter H. Cashin.

**Writing – original draft:** Lana Ghanipour, Peter H. Cashin.

**Writing – review & editing:** Lana Ghanipour, Gabriella Jansson Palmer, Per J. Nilsson, Caroline Nordenvall, Jan-Erik Frödin, Elinor Bexe Lindskog, Dan Asplund, Torbjörn Swartling, Wilhelm Graf, Helgi Birgisson, Ingvar Syk, Victor Verwaal, Jenny Brändstedt.

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
