## [Decision Letter · Decision Letter 0]

14 Aug 2023

PONE-D-23-17095Efficacy of Hyperthermic Intraperitoneal Chemotherapy for Peritoneal Metastases from Colorectal Cancer: a phase I and III open label randomized controlled registry-based clinical trialPLOS ONE

Dear Dr. Ghanipour,

Thank you for submitting your manuscript to PLOS ONE. After careful consideration, we feel that it has merit but does not fully meet PLOS ONE’s publication criteria as it currently stands. Therefore, we invite you to submit a revised version of the manuscript that addresses the points raised during the review process.

I thank the authors for the opportunity to evaluate their research protocol, which I personally considered very interesting.

I believe that the reviewers did an excellent job of identifying weaknesses in the protocol, providing the appropriate guidance so that the authors could revise and strengthen an already well-done paper.

Reviewer 1 has electively addressed the clinical issue, and I think the answers to his questions are important to make the research proposal interesting to those who are involved in the clinical pathway of patients with peritoneal metastases.

Reviewer 2 asked statistical questions: the answers may strengthen the value of the research

We look forward to receiving your revised manuscript.

Kind regards,

Fabrizio D'Acapito, Ph.D,M.D.

Academic Editor

PLOS ONE

“Complete funding has been achieved through the Swedish research council.”

6. We note that the original protocol file you uploaded contains a confidentiality notice indicating that the protocol may not be shared publicly or be published. Please note, however, that the PLOS Editorial Policy requires that the original protocol be published alongside your manuscript in the event of acceptance. Please note that should your paper be accepted, all content including the protocol will be published under the Creative Commons Attribution (CC BY) 4.0 license, which means that it will be freely available online, and any third party is permitted to access, download, copy, distribute, and use these materials in any way, even commercially, with proper attribution.

Therefore, we ask that you please seek permission from the study sponsor or body imposing the restriction on sharing this document to publish this protocol under CC BY 4.0 if your work is accepted. We kindly ask that you upload a formal statement signed by an institutional representative clarifying whether you will be able to comply with this policy. Additionally, please upload a clean copy of the protocol with the confidentiality notice (and any copyrighted institutional logos or signatures) removed.

Additional Editor Comments:

I thank the authors for the opportunity to evaluate their research protocol, which I personally considered very interesting.

I believe that the reviewers did an excellent job of identifying weaknesses in the protocol, providing the appropriate guidance so that the authors could revise and strengthen an already well-done paper.

Reviewer 1 has electively addressed the clinical issue, and I think the answers to his questions are important to make the research proposal interesting to those who are involved in the clinical pathway of patients with peritoneal metastases.

Reviewer 2 asked statistical questions: the answers may strengthen the value of the research

Reviewers' comments:

Reviewer's Responses to Questions

**Comments to the Author**

1. Does the manuscript provide a valid rationale for the proposed study, with clearly identified and justified research questions?

Reviewer #1: Partly

Reviewer #2: Yes

2. Is the protocol technically sound and planned in a manner that will lead to a meaningful outcome and allow testing the stated hypotheses?

Reviewer #1: Partly

Reviewer #2: Partly

3. Is the methodology feasible and described in sufficient detail to allow the work to be replicable?

Reviewer #1: No

Reviewer #2: No

4. Have the authors described where all data underlying the findings will be made available when the study is complete?

Reviewer #1: Yes

Reviewer #2: Yes

5. Is the manuscript presented in an intelligible fashion and written in standard English?

Reviewer #1: Yes

Reviewer #2: Yes

6. Review Comments to the Author

You may also provide optional suggestions and comments to authors that they might find helpful in planning their study.

Reviewer #1: I have read with interest the manuscript entitled: Efficacy of Hyperthermic Intraperitoneal Chemotherapy for Peritoneal Metastases from Colorectal Cancer: a phase I and III open label randomized controlled registry-based clinical trial. However, some issues must be solved before considering for publication even in a study protocol:

- The standard treatment for colorectal peritoneal metastases is not CRS and HIPEC. Authors cannot include as standard arm HIPEC with Ox 460 mg/m2, even when this treatment did not demonstrate benefit but an increasement of morbidity.

- The phase I study has been well designed.

- Phase III trial must be revised. As I referred before the CRS and HIPEC is not the standard treatment for peritoneal metastases. Maybe authors could include the cytoreduction for the management of these, but no HIPEC.

- Systemic chemotherapy must be specified.

- The use of immunotherapy must be included in analysis.

- RAS/RAF mutations must be included, even for stratification.

- The primary endpoint must be better defined. Is recurrence free survival different of disease free survival. Time to recurrence (any) or death (any cause) in months.

- Authors have established a 50% DFS at 12 months in the control arm, what have been the references for this data?. They want to increase DFS from 50% up to 66% in only 12 months when the most of recurrences appear at 18-24 months after treatment. Have authors calculated how many events they will need to get this difference at 12 months?

- Authors must explain better the follow up of the patients, since the primary endpoint is at 12 months, only two CT scans will be planned 6 months each to detect early recurrences. Will PET scan or liver MRI or biopsy or tumor marker raising be considered to define recurrences?.

- The primary endpoint comparison must be using Log-rank test and KM curves for survival endpoints. Cox regression must be planned using more risk factors that could modified the results as tumor location, RAS/RAF mutations, MMI, systemic chemo, use of antiEGFR or anti-VEGF.

- Authors have explained why patients with peritoneal metastases and liver metastases do not receive systemic chemo as neoadjuvant.

- Type of recurrence must be recorded.

- QoL questionaries at 36 months are not needed. The addition of EPIC could modify the QoL in the early postoperative, I have doubts about any impact in QoL more than one month.

- Mitomycin C has demonstrated efficacy in recent published HIPECT4 trial, it must be discussed.

Reviewer #2: The authors present the protocol for their phase I/phase III study of a HIPEC/EPIC regimen in patients with peritoneal metastases from colorectal cancer that is being conducted in Sweden. The study is reasonably described, although there are some details missing and some places that would benefit from clarification. Specifically, the protocol manuscript will be strengthened if the authors consider the following points.

1. If the loss to follow-up is 10%, the actual sample size will be slightly smaller than what their power calculation suggests is needed (90% of 163 is 146-147 patients not 148).

2. Are there accepted levels of kidney, liver, and bone marrow function based on lab tests? To provide a replicable protocol, authors should detail what is being measured and what "adequate levels" mean.

3. In the statistical analysis section, authors define a stopping rule for the phase III study based on an interim analysis as a score (on the CD scale) higher than 3 in 40% of included patients - should this be 40% or more of patients?

4. They further state that an overall survival analysis will be performed as part of the interim analysis - how much follow-up will there be on these patients when the interim analysis is conducted?

5. Authors describe a logistic regression as a secondary analysis for the Phase III study, though it is not clear what this will add to the primary analysis that uses Cox regression.

6. Authors also mention a Fisher's exact test and a Kaplan-Meier curve with the log-rank test - are these considered secondary analyses? These seem more descriptive than anything else. (also note that the sentence describing this in the manuscript is an incomplete sentence and should be edited).

7. It is not clear why authors would consider changing the primary outcome after the interim analysis. What criteria will be used to make this determination? Why change the outcome?

8. What is the justification for using different block randomization strategies for those with and without liver metastases?

9. A minor edit - in the Timeline section, authors say "Maj" for when the study began - I'm assuming this should be "May".

7. PLOS authors have the option to publish the peer review history of their article (what does this mean?). If published, this will include your full peer review and any attached files.

Reviewer #1: No

Reviewer #2: No

---

## [Author Response · Author response to Decision Letter 0]

20 Sep 2023

Thank you for your comments.

Our answers are presented below. 

1. The standard treatment for colorectal peritoneal metastases is not CRS and HIPEC. Authors cannot include as standard arm HIPEC with Ox 460 mg/m2, even when this treatment did not demonstrate benefit but an increasement of morbidity.

a. This is a statement that can be challenged, and there are different opinions regarding the truth in this statement. Since 2003, the Swedish standard treatment has been CRS and HIPEC with oxaliplatin. One single study, with several limitations, has not been considered enough to change standard praxis in Sweden. The PRODIGE 7 study made a great effort to answer the question regarding the benefit of HIPEC. However, it had some significant issues that have been brought up in several communications and although the space here does not allow a full critique, a few issues deserve being pointed out. The 1-year recurrence free survival difference was 46% in the no-HIPEC arm and 59% in the HIPEC arm, yielding a p-value of 0.06 in the limited size population included in PRODIGE 7. Furthermore, several subgroups did show a difference. Namely, the PCI 10-15 group and the no-preoperative chemotherapy group (only 13% did not receive preoperative chemotherapy). 

b. To include a no-HIPEC arm was debated greatly within the Swedish Peritoneal Oncology Group. Having no-HIPEC as standard would imply a change to preoperative chemotherapy (as in PRODIGE 7) and we have found no solid evidence for such an approach. In Sweden, upfront CRS+HIPEC is that standard of care, with good results, and there is not a long-time gap between planning for surgery and surgery in itself. Hence, there has been no need for neoadjuvant treatment while waiting for surgery. Changing the whole algorithm to include preoperative chemotherapy as standard of care would take convincing the medical oncologist community about its (unproven) benefit. No RCT have been conducted and the retrospective studies are suboptimal, and most have not shown any benefit at all.

c. We were unable to change the Swedish standard of care. Thus, to this day, the standard of care is upfront CRS+HIPEC using oxaliplatin 460mg/m2. We did consider changing to mitomycin C. However, a really large study is currently finishing a third review with 2000 patients from 39 HIPEC centers comparing oxaliplatin HIPEC with mitomycin HIPEC in the colorectal cancer setting and oxaliplatin performs better in all subgroups.

d. Long answer to this question since it is so central to this trial. We kept the oxaliplatin 30 min HIPEC as the control arm because upfront CRS+HIPEC with this regimen is currently the Swedish standard of care still. We did discuss this again at a recent study committee meeting due to this review. However, the consensus was to keep the design unchanged.

2. The phase I study has been well designed.

a. Thank you for this encouraging comment.

3. Phase III trial must be revised. As I referred before the CRS and HIPEC is not the standard treatment for peritoneal metastases. Maybe authors could include the cytoreduction for the management of these, but no HIPEC.

a. Same comment as above – please see the extensive response to comment 1.

4. Systemic chemotherapy must be specified.

a. Since, preoperative chemotherapy is not standard of care. This is not specified. Some patients will receive neoadjuvant chemotherapy for downstaging. However, the use of this is limited and conversion therapy is therefore not specified. 

b. Adjuvant chemotherapy is given according to national guidelines and standard of care in Sweden. Basically, it entails giving adjuvant therapy if the primary tumor indicates adjuvant therapy in the synchronous setting. In Sweden, we do not in general give adjuvant therapy to metachronous peritoneal metastases. The standard of care has not been included in the protocol, but since most of the patients receive their adjuvant therapy outside a study center, it was not feasible to require treatment outside the Swedish standard of care. 

5. The use of immunotherapy must be included in analysis.

a. The use of immunotherapy prior to CRS and HIPEC will be registered. Unfortunately, it is beyond the scope of the trial to follow up all chemo/immunotherapy given unto death. The primary endpoint is time to first recurrence and so treatments given after first recurrence will not affect the primary endpoint at all. Thus, all chemotherapy administered prior to CRS+HIPEC and given as adjuvant chemotherapy will be recorded for full exploratory analysis.

6. RAS/RAF mutations must be included, even for stratification.

a. We will be collecting RAS/RAF mutations as well as MMR status. However, we will not include this status in randomization stratifications. Statisticians differ on whether stratifications are good or not. Some very essential stratifications may be reasonable. We have chosen treatment center, presence of liver metastases and PCI for stratification. Adding molecular biomarkers to this will overcrowd the stratification. 

7. The primary endpoint must be better defined. Is recurrence free survival different of disease free survival. Time to recurrence (any) or death (any cause) in months.

a. The difference between DFS and RFS is small. We have added the definition of RFS to the trial for clarity: “The consensus definition of RFS includes all causes of death, anastomotic relapse and metastatic relapse as an event, but not second primary CRCs nor second primary non-CRC.” The difference compared to DFS is that second primary CRC is regarded as an event. 

8. Authors have established a 50% DFS at 12 months in the control arm, what have been the references for this data? They want to increase DFS from 50% up to 66% in only 12 months when the most of recurrences appear at 18-24 months after treatment. Have authors calculated how many events they will need to get this difference at 12 months?

a. The median time to recurrence is 12 months not 18-24 months. In the PRODIGE 7 study with a similar design, the median DFS was 11 for CRS without HIPEC and 13 with CRS and HIPEC. In fact, these figures are very consistent in many studies. Overall, it appears that approximately 50% of patients recur within 12 months, an observation that has remained unaltered over 20 years. The RCT by Verwaal et al. from 2003 had a median PFS of 12.6 months. The projected or estimated 66% RFS rate was derived from an unpublished cohort where there was a small subgroup having received the combination oxaliplatin + irinotecan + early postoperative intraperitoneal chemotherapy (EPIC). However, this cohort has now been published: Cashin PH, Esquivel J, Larsen SG, Liauw W, Alzahrani NA, Morris DL, Kepenekian V, Sourrouille I, Dumont F, Tuech JJ, Ceribelli C, Doussot B, Sgarbura O, Quenet F, Glehen O, Fisher OM; Peritoneal Surface Oncology Group International (PSOGI); Nordic Peritoneal Oncology Group (NPOG); American Society for Peritoneal Surface Malignancy (ASPSM); BIG-RENAPE Groups. Perioperative chemotherapy in colorectal cancer with peritoneal metastases: A global propensity score matched study. EClinicalMedicine. 2022 Nov 24;55:101746. 

Additionally, a second report investigating different HIPEC regimens in this 2000 patient cohort is under review in BJS Open currently. We have calculated the simple size to be able to prove a difference in RFS between 50% and 66%, i.e., a 16% increase in recurrence free survival at 12 months.

9. Authors must explain better the follow up of the patients, since the primary endpoint is at 12 months, only two CT scans will be planned 6 months each to detect early recurrences. Will PET scan or liver MRI or biopsy or tumor marker raising be considered to define recurrences?

a. We will be evaluating recurrences according to RECIST criteria where definitions of new lesions are clarified. There will be 2 CT scans available for this analysis. However, in practice we know that these patients are commonly scanned more often than at 6 and 12 months. This is due to many reasons related to complications or adjuvant therapy use or serum tumor marker increases. However, all patients will be follow-up up at a minimum of 6 months and 12 months.

b. As explain above, RECIST criteria (please refer to RECISTS 1.1 guidelines) allow for PET scan and MRI to aid in determining if a new lesion is considered a recurrence. Biopsy is not included in the protocol and neither is serum tumor markers. However, if taken outside the regular intervals, an elevated tumor marker will be followed by a CT scan. Still, it is the CT scan date that is used for recurrence free survival.

10. The primary endpoint comparison must be using Log-rank test and KM curves for survival endpoints. Cox regression must be planned using more risk factors that could modified the results as tumor location, RAS/RAF mutations, MMI, systemic chemo, use of antiEGFR or anti-VEGF.

a. Log-rank and KM generally requires much longer follow-up and overall survival analyses require MANY more patients to prove a difference. 

b. The purpose of this trial is a “proof of concept”. The aim is to investigate an intensified HIPEC schedule with regards to efficacy on RFS and toxicity. The trial design will not be able to show any potential OS gains. But, in all truth, this is what we need to be aiming for in the future-larger collaborations over several countries.

11. Authors have explained why patients with peritoneal metastases and liver metastases do not receive systemic chemo as neoadjuvant.

a. The inclusions of only 3 small technically uncomplicated liver metastases are allowed. In accordance with, e.g. ESMO guidelines, neoadjuvant therapy is not mandatory in this situation and its use is currently low, and declining, in Sweden. Hence, this trial protocol allows for a pragmatic approach with high acceptance in the oncological and HPB surgery communities in Sweden. 

12. Type of recurrence must be recorded.

a. We thank the reviewer for pointing this out and agree that this is important. This parameter is already included in the eCRF forms, and we have added this into the manuscript. 

13. QoL questionaries at 36 months are not needed. The addition of EPIC could modify the QoL in the early postoperative, I have doubts about any impact in QoL more than one month.

a. It is true that it is not reasonable to think that the difference in HIPEC regimen in this trial will affect QoL after 3 years. However, the QoL is part of a long-term follow-up of all of our HIPEC patients including other diagnoses (i.e. pseudomyxoma peritonei). So, these patients will be included in this follow-up since it is part of our clinical routine. We have opted to keep them in this routine to make it easier. 

14. Mitomycin C has demonstrated efficacy in recent published HIPECT4 trial, it must be discussed.

a. We have added a short discussion concerning this trial. Please see the discussion section. It is a positive trial, also using the same locoregional endpoint as this current trial. The indication is of course different, but still relevant to include in this discussion.

Reviewer 2

1. If the loss to follow-up is 10%, the actual sample size will be slightly smaller than what their power calculation suggests is needed (90% of 163 is 146-147 patients not 148).

a. Thank you for this help. We will adjust this in the manuscript

2. Are there accepted levels of kidney, liver, and bone marrow function based on lab tests? To provide a replicable protocol, authors should detail what is being measured and what "adequate levels" mean.

a. Thank you for this comment. We have added a short clarification. “Lab test needs to be normal according to lab reference values or within +/- 20% of the highest respective lowest value.

3. In the statistical analysis section, authors define a stopping rule for the phase III study based on an interim analysis as a score (on the CD scale) higher than 3 in 40% of included patients - should this be 40% or more of patients?

a. Yes, it should be 40% or more. This has been adjusted.

4. They further state that an overall survival analysis will be performed as part of the interim analysis - how much follow-up will there be on these patients when the interim analysis is conducted?

a. Thank you for this comment. We have missed to define this clearly. We have decided to define a minimum of 12 months follow-up. This has been added to the manuscript.

5. Authors describe a logistic regression as a secondary analysis for the Phase III study, though it is not clear what this will add to the primary analysis that uses Cox regression.

a. This is a correct remark from the reviewer and the logistic regression does not add anything. In a previous protocol version logistical regression was first defined but since it seemed inappropriate this was incorporated. However, logistic regression has not been removed from the current protocol version. 

6. Authors also mention a Fisher's exact test and a Kaplan-Meier curve with the log-rank test - are these considered secondary analyses? These seem more descriptive than anything else. (also note that the sentence describing this in the manuscript is an incomplete sentence and should be edited).

a. Correct, these are secondary analyses as written that are in some ways more descriptive. The sentence has been edited “A Fisher’s exact test for 12-months recurrence free survival between the arms, a Kaplan-Meier curve with two-tailed log rank test between the arms with recurrence free survival up to 5 years as endpoint (i.e. time to recurrence) will be performed.”

7. It is not clear why authors would consider changing the primary outcome after the interim analysis. What criteria will be used to make this determination? Why change the outcome?

a. We are inclined to agree that it may wishfully thinking. However, it is probably not reasonable to expect overall survival, which is the stronger endpoint, to be achievable with this trial size. The idea is to leave a possibility, however small, to increase the trial size in case it appears reasonable to achieve overall survival difference by a modest increase in sample size. 

8. What is the justification for using different block randomization strategies for those with and without liver metastases?

a. The combined treatment of liver and peritoneal metastases is less common. The original idea was to discuss these patients in a national conference to keep the indications strict in this smaller subset. Thus, it was a “national” inclusion situation for patients with liver engagement whereby randomization of these patients follow a separate track. 

9. A minor edit - in the Timeline section, authors say "Maj" for when the study began - I'm assuming this should be "May".

a. Thank you for this comment. This has been adjusted.

With best regards

Lana Ghanipour

---

## [Decision Letter · Decision Letter 1]

25 Oct 2023

Efficacy of Hyperthermic Intraperitoneal Chemotherapy in Colorectal Cancer: a phase I and III open label randomized controlled registry-based clinical trial protocol

PONE-D-23-17095R1

Dear Dr. Ghanipour,

We’re pleased to inform you that your manuscript has been judged scientifically suitable for publication and will be formally accepted for publication once it meets all outstanding technical requirements.

Kind regards,

Fabrizio D'Acapito, Ph.D,M.D.

Academic Editor

PLOS ONE

Additional Editor Comments (optional):

I congratulate the authors on the quality of their paper.

I believe that all the annotations made by the reviewers were properly evaluated and adequately included in the revision of the article.

Reviewers' comments:

Reviewer's Responses to Questions

**Comments to the Author**

1. Does the manuscript provide a valid rationale for the proposed study, with clearly identified and justified research questions?

Reviewer #2: Yes

2. Is the protocol technically sound and planned in a manner that will lead to a meaningful outcome and allow testing the stated hypotheses?

Reviewer #2: Yes

3. Is the methodology feasible and described in sufficient detail to allow the work to be replicable?

Reviewer #2: Yes

4. Have the authors described where all data underlying the findings will be made available when the study is complete?

Reviewer #2: Yes

5. Is the manuscript presented in an intelligible fashion and written in standard English?

Reviewer #2: Yes

6. Review Comments to the Author

You may also provide optional suggestions and comments to authors that they might find helpful in planning their study.

Reviewer #2: The authors thoughtfully addressed all of my earlier concerns, so I have nothing further to add to the review.

7. PLOS authors have the option to publish the peer review history of their article (what does this mean?). If published, this will include your full peer review and any attached files.

Reviewer #2: No

---

## [Editor Report · Acceptance letter]

13 Nov 2023

PONE-D-23-17095R1 

Efficacy of Hyperthermic Intraperitoneal Chemotherapy in Colorectal Cancer: a phase I and III open label randomized controlled registry-based clinical trial protocol 

Dear Dr. Ghanipour:

I'm pleased to inform you that your manuscript has been deemed suitable for publication in PLOS ONE. Congratulations! Your manuscript is now with our production department. 

Kind regards, 

on behalf of

Dr. Fabrizio D'Acapito 

Academic Editor

PLOS ONE